# Nutritional, Productive, and Qualitative Characteristics of European Quails Fed with Diets Containing Lipid Sources of Plant and Animal Origin

**DOI:** 10.3390/ani13223472

**Published:** 2023-11-10

**Authors:** Jean Kaique Valentim, Rodrigo Garófallo Garcia, Maria Fernanda de Castro Burbarelli, Rosa Aparecida Reis de Léo, Rafael de Sousa Ferreira, Arele Arlindo Calderano, Ideraldo Luiz Lima, Karina Márcia Ribeiro de Souza Nascimento, Thiago Rodrigues da Silva, Luiz Fernando Teixeira Albino

**Affiliations:** 1Animal Science Postgraduate Program, Federal University of Viçosa (UFV), Viçosa 36570-900, MG, Brazil; rosa.leo@ufv.br (R.A.R.d.L.); rafael.d.ferreira@ufv.br (R.d.S.F.); calderano@ufv.br (A.A.C.); ideraldo.lima@gmail.com (I.L.L.); lalbino@ufv.br (L.F.T.A.); 2Animal Science Postgraduate Program, Federal University of Grande Dourados (UFGD), Dourados 79804-970, MS, Brazil; rodrigogarcia@ufgd.edu.br (R.G.G.); fariakita@gmail.com (M.F.d.C.B.); 3Animal Science Postgraduate Program, Federal University of Mato Grosso do Sul (UFMS), Campo Grande 79070-900, MS, Brazil; karina.souza@ufms.br (K.M.R.d.S.N.); thiagoth_rodrigues@hotmail.com (T.R.d.S.)

**Keywords:** alternative oils, metabolizable energy, metabolism, meat quality, non-conventional lipid

## Abstract

**Simple Summary:**

This study focused on evaluating the impact of different alternative lipid sources in the diet of meat quails. The researchers investigated nutrient metabolism, performance, carcass characteristics, and meat quality. Two experiments were conducted, the first of which analyzed the energy values and metabolizability of various lipid sources. Surprisingly, no significant differences were observed in energy values and metabolizability among the different lipid sources. However, variations were found in the metabolizability of crude protein and ether extract. Based on these findings, the second experiment assessed performance, carcass characteristics, and meat quality. Notably, the inclusion of distilled corn oil had positive effects on carcass yield and skin/meat color parameters. This study demonstrates the successful incorporation of alternative lipid sources into the diet of meat quails, with no compromise to performance and carcass characteristics. These findings provide valuable insights into improving nutrition and meat quality, benefiting both the quail industry and society as a whole.

**Abstract:**

The objective of this study was to investigate the impact of alternative lipid sources on nutrient metabolism, performance, carcass characteristics, and meat quality in European quails. Trial 1 determined the energy values and nutrient metabolizability of non-conventional lipid sources. Six treatments (control, soybean oil, conventional corn oil, distilled corn oil, poultry fat, and beef tallow) were randomly assigned with 10 replicates per treatment. Trial 2 evaluated animal performance, carcass characteristics, and meat quality using a randomized design with five treatments and 10 replicates each. Statistical analysis showed no significant difference in apparent metabolizable energy corrected by nitrogen (AMEn) and coefficients of metabolizability (CM%) among the lipid sources. The AMEn values found were 8554 for soybean oil, 7701 for corn, 7937 for distilled corn oil, 7906 for poultry fat, and 7776 for beef tallow (kcal/kg). The CM values were 88.01% for soybean oil, 79.01% for corn oil, 84.10% for distilled corn oil, 81.43% for poultry fat, and 79.28% for beef tallow. The inclusion of lipid sources of plant and animal origin in the diet of 7–35-day old meat quails did not influence performance or carcass and cut characteristics. The inclusion of distilled corn oil increased carcass yield and influenced skin and meat color parameters. AMEn values varied for each lipid source. The inclusion of distilled corn oil positively influenced skin and meat color as well as carcass yield in quails.

## 1. Introduction

In poultry production, nutrition occupies a prominent place since feeding is one of the costliest factors in breeding. According to [1,2], among the nutrients necessary for bird development, lipids stand out as sources of fatty acids that act in various metabolic functions of the organism. Nutrient adequacy, such as fatty acids for birds, can occur traditionally through the diet provided and be formulated according to recommendations for each animal phase by adding lipid sources rich in fatty acids derived from vegetable oils or animal fat [3].

The energy of the diet is among the most relevant points in the formulation of poultry feed, representing the sum of all nutrients and influencing performance and carcass characteristics [4,5]. In practice, nutritionists must use vegetable oils and/or animal fat to achieve the appropriate level of energy in bird diets [6,7]. Oils and fats not only contribute to the energy of the diet but also represent a range of ingredients that provide considerable portions of important nutrients for proper nutritional balance, providing the substrates necessary for the proper functioning of the organism [8,9].

Lipids are chemically formed by the association of one, two, or three molecules of fatty acids, which are classified as saturated when they have only single bonds between carbons, and unsaturated when they have one or more double bonds in their chain [10]. These differences are mainly related to their origin, whether vegetable or animal, and this determines their degree of digestibility in the animal body [11].

Soybean oil is the main option for increasing the energy density of feed, which is obtained through the extraction and degumming of raw soybean oil [12], presenting the main advantage of low production costs and high metabolizable energy content (8331 kcal/kg) [13]. The author of [14] highlights those unconventional sources, such as soy or corn acid oils and animal fat sources, are less costly and attractive alternatives for use in animal diets, as they have the same profile as degummed soybean oil, being rich in polyunsaturated fatty acids.

Animal fats are commonly used in the nutrition of non-ruminant animals and are derived from the carcass residues of birds, fish, cattle, and pigs [15]. Therefore, due to the large variations, it is necessary to know the individual actions of the various lipid sources in quail feeding [16], given the lack of content in the current literature.

Experiments which evaluate the metabolic utilization of foods, such as that proposed in this research, are necessary and periodically required for updating the nutritional requirements of birds, mainly due to the constant changes in the feed efficiency of quails which alter the use of nutrients in their diet [15]. 

Regional issues can also interfere with ingredient quality, which varies depending on climate, soil, planting conditions, cultivars used, and food storage and processing methods [17]. In addition to the few data in the literature, the discrepancies observed between published reports are probably caused by variations in the nutritional requirements used as a basis for diet formulation.

The main importance of using lipid sources in poultry diets lies in the fatty acids that act in physiological processes of the organism, such as energy storage, maintenance of structure, and integrity, and affect the functioning of plasma membranes, improve the organism’s immunity, and influence the meat quality [18,19]. In this respect, different lipid sources can act as enhancers of poultry diets, aiming to improve the fatty acid composition of the meat and yolk, as well as the health and performance of the birds [20].

Thus, this research aimed to evaluate lipid sources of plant and animal origin in the diet of meat quails, seeking to determine their energy values and nutrient metabolizability, and their influence on performance, carcass and organ characteristics, and meat quality.

## 2. Materials and Methods

The two experiments were conducted at the Teaching, Research, and Extension Unit in Poultry Production and Nutrition of the Department of Animal Science at the Agricultural Sciences Center of the Federal University of Viçosa, Viçosa, Minas Gerais, Brazil (20°45′57.19″ S, 42°51′35.42″ W; and 682 m above sea level).

In Experiment 1, the energy values and nutrient metabolizability of lipid sources of plant and animal origin in the diet of European quails were determined. Six treatments were used (control, soybean oil, conventional corn oil, distilled corn oil, poultry fat, and beef tallow) in a completely randomized design with 10 replicates per treatment, with eight quails per experimental unit, totaling 480 European quails (Coturnix coturnix coturnix), of both sexes, of 21 days of age, and an average weight of 130.35 ± 0.025 g. The basal diet of this experiment was composed of corn and soybean meal, without the inclusion of lipid sources and according to the food composition and nutritional requirements established by [21].

The birds were acquired at one day of age and raised until 21 days in protection circles, equipped with heating bells, drinkers, and infant feeders, correctly dimensioned for the number of animals used. After 21 days, the quails were transferred to metabolic cages with two-story compartments arranged in a 68 m^2^ room with a ceiling height of approximately 2.8 m. The metabolic cages were constructed of metal with dimensions of 50 × 50 × 16.5 cm (length × width × height), totaling 1250 cm², equipped with nipple drinkers with cups and trough feeders positioned at the front of the cages. The evaluation of the lipid sources was performed by the total collection method of excreta when the birds reached 22 days of age. 

The test diets were obtained by substituting 8% of the basal diet with the lipid source, where according to [22] the best level for this type of digestibility assay is 8–10%, with the level of 8% chosen to avoid possible hepatic overload of the birds. The experimental diets consisted of: T1—basal diet; T2—basal diet + 8% soybean oil inclusion; T3—basal diet + 8% conventional corn oil inclusion; T4—basal diet + 8% distilled corn oil; T5—basal diet + 8% poultry fat; T6—basal diet + 8% beef tallow (Table 1). 

The soybean and conventional corn oils were used in refined form, while distilled corn oil, poultry fat, and beef tallow were used in crude form, all obtained commercially. Water and feed were provided ad libitum. For the identification of fatty acids in rations, samples were collected, homogenized, and frozen at −20 °C to preserve the characteristics of volatile fatty acids. 

After the esters were obtained, they were analyzed on a Shimadzu GC-17 gas chromatograph equipped with a flame ionization detector with manual injection, a capillary column (CARBOWAX), and H2 as the carrier gas. The calculations were performed by integration with a computer connected to the detector, extracting the total amount of fatty acids from the diets, as well as the percentage of saturated and unsaturated fatty acids.

Trial 1. Energy values and nutrient metabolizability: The evaluation of the AMEn of lipid sources was carried out using the method of total excreta collection and an iron oxide marker when the poultry reached nine weeks of age. The metabolic assay began five days after the provision of experimental diets, followed by five days when excreta were collected. All cages were equipped with a tray previously prepared for the collection of excreta, which was cleaned out twice a day at 8:00 am and 5:00 pm.

The excreta were placed in plastic bags identified by replication and stored in a freezer at −16 °C. At the end of the collection period, the amount of feed consumed and the total amount of excreta produced were determined. At the time of analysis, the samples were defrosted and homogenized. 

An aliquot of about 200 g of excreta from each replication was removed and weighed and then placed in an oven with forced air ventilation at 55 °C for 72 h to proceed with pre-drying. Subsequently, the samples were exposed to air to equilibrate with ambient temperature and humidity. Then, they were weighed, ground in 1-mm knife mills, and placed in containers for laboratory analysis of dry matter (DM), ether extract (EE), crude protein (CP), AMEn /kcal/kg, and mineral matter (MM) according to [23].

The moisture and nitrogen contents of the excreta and rations were determined according to the methodology described by [23]. The gross energy of diets, lipid sources, and excreta was obtained using a calorimetric bomb (IKA^®^ model PARR 6200). The metabolizable energy (ME) and AMEn values were calculated using the equations proposed by [24]:$$\mathrm{ME}\;\mathrm{or}\;\mathrm{BF}\;\left(\mathrm{kcal}/\mathrm{kg}\right)=\frac{\left(\mathrm{GE}\;\mathrm{ingested}-\mathrm{GE}\;\mathrm{excreted}\right)}{\mathrm{Feed}\;\mathrm{intake}}$$
$$\mathrm{ME}\;\mathrm{of}\;\mathrm{lipid}\;\mathrm{source}\;\left(\mathrm{kcal}/\mathrm{kg}\right)=\mathrm{ME}\;\mathrm{BF}\;+\;\frac{\left(\mathrm{ME}-\mathrm{ME}\;\mathrm{BF}\right)}{\%\;\mathrm{of}\;\mathrm{replacement}}$$
$$\mathrm{AMEn}\;or\;\mathrm{BF}\;\left(\mathrm{kcal}/\mathrm{kg}\right)=\frac{(\mathrm{GE}\;\mathrm{ingested}-(\mathrm{GE}\;\mathrm{excreted}+8.22\times \mathrm{NB}))}{\mathrm{Feed}\;\mathrm{intake}}$$
where: BF = basal feed; GE = gross energy; NB = nitrogen balance (N ingested − N excreted).

The calculation of the AMEn metabolizability coefficient (CM AMEn) was obtained by the ratio between MEn and GE ingested and excreted and expressed as a percentage:$$\mathrm{AMEn}\;\mathrm{of}\;\mathrm{lipid}\;\mathrm{source}\;\left(\mathrm{kcal}/\mathrm{kg}\right)=\mathrm{AMEn}\;\mathrm{BF}+\frac{\left(\mathrm{MEn}-\mathrm{MEn}\;\mathrm{BF}\right)}{\%\;\mathrm{of}\;\mathrm{replacement}}$$
$$CM\;AMEn\;\left(\%\right)=\frac{AMEn\;diet-AMEn\;excreta}{AMEn\;diet\times 100}$$

Trial 2. Performance, carcass characteristics, and meat quality: Based on the energy values and MC found for each lipid source, this information was used to formulate the diets used in Experiment 2 to evaluate performance, carcass yield, and meat quality. This experiment was carried out in a completely randomized design with five treatments (soybean oil, conventional corn oil, distilled corn oil, poultry fat, and beef tallow) with 10 replicates each and 12 European quails of the Fujikura commercial lineage (Coturnix coturnix coturnix), with seven days per experimental unit, totaling 600 birds.

The birds were housed in galvanized wire cages, made of perforated metal material and with walls of 108 cm with dimensions of 50 × 50 × 16.5 cm (length × width × height), totaling 1250 cm². The cages were arranged on 120 cm high masonry benches, equipped with nipple-type drinkers with cups and trough-type feeders, positioned at the front of the cages, and a masonry floor covered with 4 cm thick sawdust bedding. The animal density per experimental unit was 156 cm²/bird. Water and experimental diets were provided ad libitum.

The diets were composed of corn and soybean meal and calculated according to the food composition and nutritional requirements established by [16] for the 7–21-day (Table 2) and 22–35-day (Table 3) phases.

Daily management consisted of providing feed and monitoring temperature (maximum and minimum) and relative air humidity (RH). Temperatures and RH were monitored twice a day, at 8:00 am and 4:00 pm, using maximum and minimum thermometers and dry- and wet-bulb thermometers positioned in the center of the shed at the height of the birds’ backs. The minimum temperature obtained was 18.27 ± 0.41 (°C) and the maximum was 26.9 ± 0.27 (°C), while the maximum relative humidity was 85.0 ± 2.9 (%) and the minimum was 55.5 ± 1.8 (%). The curtains were controlled by analyzing the temperature of the day.

A 24 h artificial lighting program was used from the first to the fourteenth day of the birds’ lives (raising phase), and then, from the fifteenth to the thirty-fifth day of age (slaughter age), 23 h of light (natural + artificial) were provided using a timer to promote feed intake. Incandescent bulbs of 60 watts were used for artificial lighting, with one bulb used for every four cages.

Performance evaluation: The daily weight gain per bird (WG), feed intake per bird per day (FI), and feed conversion ratio (FCR) were evaluated. According to the methodology described, WG was determined by weighing the birds on the seventh, twenty-first, and thirty-fifth days in the afternoon and calculating the average weight gain per bird per day. FCR was determined by dividing the difference between the feed provided during the phase and the leftover feed weighed at the end of the phase by the number of birds in the plot. The weighing of the leftovers was also completed using a scale, and the averages were totaled to obtain the average feed intake per bird in the plot. FCR was calculated by dividing the average feed intake by the average weight gain of the birds in the plots studied.

Yield characteristics of carcass and organs: The yield of the mixed lot was analyzed, and the weight of the carcass, breast, thigh, wings, edible viscera (heart, liver, and gizzard), and inedible viscera (intestines, reproductive organs of females, and spleen) were evaluated. The slaughter and sample collection were performed on the broilers’ thirty-fifth day. During this period, the birds were subjected to an eight-hour fast (only feed) and then two birds (one male and one female) from each plot with an average weight within the range of ±10% of the average weight of the experimental unit were selected, weighed, and identified; totaling 100 slaughtered birds.

These birds were rendered unconscious by cervical dislocation, and manual bleeding was performed by cutting the jugular vein. After mild scalding at a temperature of 56 °C for 1 min, the birds were individually plucked. After plucking, the birds were weighed again to obtain the weight of the feathers. They were then manually eviscerated, and the carcasses were placed in chillers for pre-cooling for 2 min, from which they emerged with a temperature of 8 °C. After the chiller, the carcasses were placed on a stainless-steel conveyor belt with holes for excess water drainage.

The carcasses were weighed after the removal of the head, neck, and feet using a 3 kg capacity balance. For the calculation of yield, commercial cuts were divided into breast, wings, thigh, drumstick, edible viscera (liver, gizzard, and heart), and inedible viscera (intestine and spleen).

All viscera were weighed using a semi-analytical balance (BL Series, Brand: Shimadzu, Model: BL 3200 H) with a capacity of 3200 g. To determine the weight of the gizzard, the food in the organ was removed, leaving the keratin that surrounds it. The intestine underwent slight compression to eliminate the interior content, and the clean tissue was weighed on a balance with a precision of 0.5 g.

The percentage yield of the main cuts, edible and inedible viscera, was calculated by dividing the average weight of each representative cut by the carcass weight, according to the formula: Yield x = Variable weight/Carcass weight*100.

Meat quality: For the evaluation of meat quality, the breast muscles (pectoralis major) of the slaughtered birds were kept in a cold room at 4 ± 1 °C for 24 h. After that, they were taken to the laboratory to assess their meat quality characteristics. The parameters evaluated were: pH 24h, luminosity (L*), red/green content (a*), yellow/blue content (b*) of the skin and meat, cooking loss (CL), water retention capacity (WRC), and shear force (FC).

The determination of pH 24h was performed based on [20]: using a glass-body penetration electrode at four different points of the breast muscle, two in the upper and two in the lower parts. The device used was a pH meter (Oakton, pH 300, series 35618) with automatic temperature compensation.

For the analysis of luminosity (L*), red/green content (a*), and yellow/blue content (b*), a portable colorimeter (HunterLab MiniScan EZ 45/0 LAV) was used to read the CIElab system parameters with a D65 illuminant source, calibrated on standard white porcelain with Y = 93.7, x = 0.3160, and y = 0.3323, on the breast muscle, pectoralis major, and skin removed from the breast of the birds. The final value was considered as the average of four readings obtained at different lesion-free points of the muscle, on the ventral region, two in the cranial part, and two in the central part, with the muscle and skin on an opaque surface, based on [25].

For the WRC variable, approximately 0.5 g of each breast sample was placed between two filter papers and acrylic plates, where they received a pressure exerted by a weight of 10.0 kg for 2 min. After this process, the amount of lost water absorbed by the filter paper was evaluated by the different areas. The areas obtained in the WRC results were calculated using ImageJ software. Then, WRC was expressed as the ratio between the total area of the exudate and the area under the pressed meat, according to the equation below:WRC = AT/AC

Being: WRC = Water retention capacity; PCA = Pressed meat area; TAE = Total exudate area.

To determine the CL, the weights of the chicken breasts were recorded on a precision electronic scale (Marconi^®^, model AS 1000), before and after cooking. The samples were packed in plastic bags and cooked in a water bath (Thermomix BM—18BU—Braun Biotech International), at a temperature of 80 °C for 60 min. Then, the samples were cooled for 16 h at 4 °C to perform the final weighing and evaluate the percentage of weight loss after cooking, according to [26].

After this procedure, the samples were exposed to room temperature and cut into 1 cm × 1 cm × 2 cm (height, width, and length) pieces to evaluate the meat tenderness, performed by texture analysis, which measures the force in kilograms-force (KGF) required to rupture the muscle fibers. These were placed with the fibers oriented perpendicular to the blade of the texture analyzer.

A Stable Micro Systems TAXT 2 Plus texture analyzer was used, equipped with a V Warner Bratzler probe blade set, calibrated to the standard weight of 5 kg and traceable standard. The speed of descent and cutting of the device was adjusted to 200 mm per minute. The data (maximum positive peaks) were obtained using the Exponent Lite version 5.1 program (Stable Micro Systems).

Statistical analysis: The data related to the energy metabolism of the birds were subjected to analysis of variance (PROC MIXED) and the means of the treatments were compared by Tukey’s test at a significance level of 5%. The adopted statistical model is represented as follows: *y**i* = *m* + *t**i* + *e**i*. (a × b)i
where: Yijk = response variable of the birds, which were the values of MEAn or CM (%) from different lipid sources; μ = overall effect of the mean; ai = fixed effect of treatments (lipid sources); *y**i* = *m* + *t**i* + *e**i*. (a × b)i = residual error. 

The data were analyzed using the Statistical Analysis System [27] and to verify the statistical assumptions of normality of the residuals, the Shapiro–Wilk test was used, and the homogeneity of variances was evaluated using Levene’s test. When a significant effect was observed, the means were compared using Tukey’s test at a 5% level of probability.

## 3. Results

The descriptive data on the fatty acid content of diets with different lipid sources are presented in Table 4. 

Vegetable lipid sources had a higher total and unsaturated fatty acid content, as expected, while animal lipid sources had a higher saturated fatty acid content. The values of ME (kcal/kg) found for soybean oil, corn oil, corn distillate, poultry fat, and beef tallow were 8554, 7701, 7937, 7906, and 7776 (kcal/kg), respectively. The apparent metabolizable energy coefficients were 88.01%, 79.01%, 84.10%, 81.43%, and 79.28%, respectively. There was no significant difference between lipid sources for ME (kcal/kg) and CM (%) in the evaluated diets (Table 5).

There was a significant difference (*p* < 0.05) in the coefficients of metabolizability (CM%) of CP and EE among the studied lipid sources. The diet with beef tallow had a higher MC (84.49%) compared to the soybean oil treatment (81.28%). Regarding CP CM%, the diet with poultry fat (90.94%) had a higher value than the soybean oil diet (88.69%). For the other coefficients, there was no difference between treatments (*p* > 0.05; Table 6).

The inclusion of alternative lipid sources in the diets of European quails in the 7–21 and 22–35-day phases did not have a significant effect (*p* > 0.05) on zootechnical performance variables (Table 7). 

The inclusion of distilled corn oil provided a higher carcass yield (%) (*p* < 0.05) compared to the diet containing poultry fat. For the other variables related to weight, carcass yield and evaluated cuts, there were no significant differences (Table 8). 

The treatments had no effect on the weight and yield of viscera and intestine size of the birds (Table 9). 

This is also justified by the content of total fatty acids, saturated and unsaturated fatty acids found in these sources, which were 93.00, 14.00, and 79.0 for distilled corn oil, and 87.00, 35.00, and 52.0 for poultry fat, as shown in Table 2. Significant differences were observed for skin color parameters A and B, and meat color parameters L, A, and B. For the other meat quality parameters, there were no treatment effects (Table 10). 

## 4. Discussion

The DM values of lipid sources were similar to those observed in the main nutrient composition tables [28], ranging from 99.60% to 99.80%. The values of GE (kcal/kg) were also similar, ranging from 9437 to 9809 kcal/kg of oil. The GE of corn oil (9437 kcal/g) was lower than that of the other proposed treatments, such as soybean oil (9746 kcal/g), regular corn (9437 kcal/g), corn distillate (9710 kcal/g), poultry fat (9809 kcal/g), and beef tallow (9720 kcal/g).

Corn oil is generally not widely used in animal diets since the corn grains used in feed already contain around 3.5% oil, contributing to energy availability. In this sense, breeding programs have strongly focused on cultivars that produce corn grains with up to 6% oil content [29].

The GE value of soybean oil (9746 kcal/kg) was similar to that presented by [30] (9851 kcal/kg). Regarding the coefficients of metabolizability of ether extract and crude protein in the diets, higher values were observed for bovine tallow and poultry fat, respectively. However, [8,11] do not support these findings, reporting that due to differences related to reduced absorption in the animal body with animal sources, the digestibility of animal sources may be lower when compared to vegetable sources. The non-significant difference in the performance of the birds can be justified by the accuracy in the energy supply of the diets, meeting the nutritional requirements of the birds for their growth, since the energy content of these sources was evaluated beforehand, and the diets were calculated in an isonutritive manner for all treatments.

The author of [31] also did not find any effect of replacing sunflower oil with flaxseed oil on live weight or on the average weight of thigh, breast, liver, heart, spleen, and bursa of Fabricius, which is related to meeting the energy requirements of birds, corroborating the results observed in this study. Animal-sourced fats have a lower degree of unsaturation, and therefore generally have lower digestibility [32], which may justify the better results obtained for the carcass yield of birds fed with distilled corn oil compared to poultry fat.

According to [33], meat pH is one of the main factors that affects its color as well as its physical structure, light reflectance properties, water retention capacity, tenderness, cooking weight loss, juiciness, and microbiological stability. The values found were similar to the reference values (5.6 to 5.9) recommended in the literature [25,34], indicating no interference of the diet in these aspects. There was a higher intensity of the green/red color (a) of the skin of birds fed with distilled corn sources in the diet. A lower intensity of the blue/yellow color (b) was verified for the skin of birds fed with poultry fat in the diet.

For the brightness (L) of the meat, there was a higher incidence for birds fed with soybean oil, corn oil, and poultry fat compared to birds fed with distilled corn oil and beef tallow in the diet. There was a higher intensity of the b color for meat from birds fed with poultry fat compared to the other treatments. For the color of the meat, there was a higher incidence for animals fed with conventional corn sources, distilled corn, and beef tallow.

Among sensory attributes, color has been recognized as an indicator of quality, playing an important role in food acceptance by consumers. Thus, paying attention to colorimetry parameters is important for consumer purchasing incentives in Brazil due to the appeal related to the consumption of healthy products [35]. The skin pigmentation of poultry is also an important quality of poultry meat for consumers in China, the United States, Mexico, and many other countries [36,37] due to the more intense color being associated with higher quality, vitamin content, and freshness of the products.

It is known that this characteristic is mainly linked to the carotenoid content of the feed offered to animals, which specifically in poultry, directly affects the color of egg yolk and poultry skin [38]. The positive results for higher skin and meat color of quails fed with diets containing distilled corn mainly relate to the high amount of xanthophylls in this lipid source, due to the concentration of carotenoids from corn and the distillation process of this input.

For a better comprehension of these results, Ref. [39] report that carotenoids included in diets come from liposoluble pigments synthesized by plants and photosynthetic microorganisms, and are obtained by animals through feeding. Birds, like all animals, metabolize carotenoids, but cannot synthesize them, and therefore require supplementation in their diet.

The deposition of pigments in specific tissues depends on the amount in the diet, the rate of deposition in the tissue, and the bird’s ability to digest, absorb, and metabolize them [39,40]. Free carotenoids, after being absorbed with fatty acids, are transported by lipoproteins in the blood, and after the pigments are absorbed in the ileum with fatty acids in the form of micelles, are esterified and stored mainly in adipose tissue and skin as hydroxy carotenoids [41,42]. As verified in the present study, the distilled corn source, having a higher concentration of these compounds, provided greater red coloration in the skin and meat of birds.

## 5. Conclusions

For European quails between 21 and 35 days old, it is possible to consider AMEn values for soybean oil (8554 kcal/kg), corn oil (7701 kcal/kg), distilled corn oil (7937 kcal/kg), poultry fat (7906 kcal/kg), and beef tallow (7776 kcal/kg), with MC values 88.01%, 79.01%, 84.10%, 81.43%, and 79.28%, respectively. Based on the energy values and MC found for each lipid source to formulate diets for European quails, the inclusion of lipid sources of plant and animal origin did not influence the performance of 7–35-day old meat quails. Moreover, the inclusion of distilled corn oil increased the skin and meat color and the carcass yield of birds in the phase from 7 to 35 days.

## Figures and Tables

**Table 1 animals-13-03472-t001:** Percentage and calculated composition of the experimental diets of assay 1.

Ingredients	Quantity (%)
Corn bran	49.970
Soybean meal	36.837
Limestone	1.047
Sugar	10.00
Mr.-Methionine	0.368
L-Lysine	0.234
Salt	0.373
Dicalcium Phosphate	0.967
Mineral supplement ¹	0.100
Vitamin supplement ²	0.100
Calculated nutritional composition	
Energy metabolizable (kcal/kg)	2950.000
Crude protein (CP) (%)	21.500
Digestive lysine (%)	1.230
Methionine + Cystine digestible (%)	0.940
Methionine digestible (%)	0.640
Tryptophan digestible (%)	0.230
Digestive threonine (%)	0.710
Calcium (%)	0.730
Available phosphorus (%)	0.30
Sodium (%)	0.170
Crude fiber (%)	2.740
Chlorine (%)	0.260

^1^—Mineral supplement (Amount per kg of Feed): Manganese 82 mg; Iron—58.4 mg. Selenium—0.35 mg. Copper—11.6 mg. Iodine—1.18 mg. ^2^—Vitamin supplement (Amount per kg of feed): vitamin A—13.493 IU; vitamin D3—3.374 IU. Vitamin E—50.5 IU. Vitamin B1—3.64 IU; vitamin B2—9.0 IU; vitamin B6—5.05 IU; Ac. Pantothenic—18.1 mg. Biotin—0.126 mg. Vitamin K3—2.71 mg. Folic acid—1.26 mg. Nicotinic acid—54.9 mg. Vitamin B12—0.022 mg.

**Table 2 animals-13-03472-t002:** Percentage and calculated composition of the experimental diets of assay 2 for the 7–21-day phase of the birds.

Ingredients	Lipid Sources				
	Soy	Corn	Distilled Maize	Poultry Fat	Beef Tallow
Corn bran (7.86 CP)	47.090	47.090	47.090	47.090	47.090
Soybean meal	45.557	45.557	45.557	45.557	45.557
Lipid source	2.912	3.235	3.156	3.154	3.217
Dicalcium Phosphate	1.360	1.360	1.360	1.360	1.360
Limestone	1.044	1.044	1.044	1.044	1.044
Inert	0.700	0.397	0.456	0.458	0.397
DL-Methionine	0.397	0.382	0.397	0.397	0.396
Common Salt	0.382	0.378	0.382	0.382	0.382
L-Threonine	0.218	0.218	0.218	0.218	0.218
L-Lysine	0.142	0.142	0.142	0.142	0.142
Mineral supplement ¹	0.100	0.100	0.100	0.100	0.100
Vitamin supplement ²	0.100	0.100	0.100	0.100	0.100
Calculated nutritional composition					
Energy metabolizable (kcal/kg)	2900.00	2900.00	2900.00	2900.00	2900.00
Crude protein (%)	25.00	25.00	25.00	25.00	25.00
Digestive lysine (%)	1.370	1.370	1.370	1.370	1.370
Methionine + Cystine digestible (%)	1.040	1.040	1.040	1.040	1.040
Methionine digestible (%)	0.709	0.709	0.709	0.709	0.709
Tryptophan digestible (%)	0.230	0.230	0.230	0.230	0.230
Digestive threonine (%)	1.040	1.040	1.040	1.040	1.040
Calcium (%)	0.850	0.850	0.850	0.850	0.850
Available phosphorus (%)	0.380	0.380	0.380	0.380	0.380
Sodium (%)	0.170	0.170	0.170	0.170	0.170
Crude fibre (%)	2.740	2.740	2.740	2.740	2.740
Chlorine (%)	0.278	0.278	0.278	0.278	0.278

^1^—Mineral supplement (Amount per kg of feed): Manganese 82 mg; Iron—58.4 mg. Selenium—0.35 mg. Copper—11.6 mg. Iodine—1.18 mg. ^2^—Vitamin supplement (Amount per kg of feed): vitamin A—13.493 IU; vitamin D3—3.374 IU. Vitamin E—50.5 IU. Vitamin B1—3.64 IU; vitamin B2—9.0 IU; vitamin B6—5.05 IU; Ac. Pantothenic—18.1 mg. Biotin—0.126 mg. Vitamin K3—2.71 mg. Folic acid—1.26 mg. Nicotinic acid—54.9 mg. Vitamin B12—0.022 mg.

**Table 3 animals-13-03472-t003:** Percentage and calculated composition of the experimental diets of assay 2 for the 22-35-day phase of the birds.

Ingredients	Lipid Sources				
	Soy	Corn	Distilled Maize	Poultry Fat	Beef Tallow
Corn (7.86% CP)	58.735	58.735	58.735	58.735	58.735
Soybean meal (46% CP)	35.969	35.969	35.969	35.969	35.969
Oil	1.462	1.625	1.585	1.584	1.616
Dicalcium Phosphate	1.003	1.003	1.003	1.003	1.003
Limestone	0.929	0.929	0.929	0.929	0.929
Inert	0.710	0.538	0.578	0.579	0.547
Common Salt	0.381	0.381	0.381	0.381	0.381
Mr.-Methionine	0.371	0.370	0.370	0.370	0.370
L-Lysine	0.251	0.251	0.251	0.251	0.251
Mineral supplement ¹	0.100	0.100	0.100	0.100	0.100
Vitamin supplement ²	0.100	0.100	0.100	0.100	0.100
Calculated nutritional composition					
Energy metabolizable (kcal/kg)	2950.000	2950.000	2950.000	2950.000	2950.000
Crude protein (%)	21.500	21.500	21.500	21.500	21.500
Digestive lysine (%)	1.230	1.230	1.230	1.230	1.230
Methionine + Cystine digestible. (%)	0.940	0.940	0.940	0.940	0.940
Methionine digestible (%)	0.640	0.640	0.640	0.640	0.640
Tryptophan digestible (%)	0.230	0.230	0.230	0.230	0.230
Digestive threonine (%)	0.710	0.710	0.710	0.710	0.710
Calcium (%)	0.730	0.730	0.730	0.730	0.730
Available phosphorus (%)	0.300	0.300	0.300	0.300	0.300
Sodium (%)	0.170	0.170	0.170	0.170	0.170
Crude fibre (%)	2.740	2.740	2.740	2.740	2.740
Chlorine (%)	0.260	0.260	0.260	0.260	0.260

^1^—Mineral supplement (Amount per kg of feed): Manganese 82 mg; Iron—58.4 mg; Selenium—0.35 mg; Copper—11.6 mg; Iodine—1.18 mg. ^2^—Vitamin supplement (Amount per kg of feed): vitamin A—13.493 IU; vitamin D3—3.374 IU; Vitamin E—50.5 IU; Vitamin B1—3.64 IU; vitamin B2—9.0 IU; vitamin B6—5.05 IU; Ac. Pantothenic—18.1 mg; Biotin—0.126 mg; Vitamin K3—2.71 mg; Folic acid—1.26 mg; Nicotinic acid—54.9 mg; Vitamin B12—0.022 mg.

**Table 4 animals-13-03472-t004:** Descriptive analysis of the fatty acid content of diets containing different lipid sources used.

Fatty Acid Profile (%)	Lipid Sources				
	Soy	Corn	Distilled Maize	Poultry Fat	Beef Tallow
Total fatty acids	90.5	90.8	93.00	87.00	85.00
Saturated fatty acids	15.0	12.8	14.00	35.00	38.00
Unsaturated fatty acids	75.5	78.0	79.0	52.0	47.0
C16:0 (Palmitic)	14.103	16.550	14.048	20.786	20.169
C16:1 (Palmitoleico)	0.536	0.596	0.677	7.016	5.536
C18:0 (Stearic)	7.208	6.036	4.175	8.118	6.197
C18:1w9 (Oleic)	26.963	35.011	56.424	43.008	37.075
C18:2w6 (Linoleic)	45.313	36.912	24.618	17.157	19.863
C18:3w3 (Linolenic)	2.143	1.132	1.132	0.900	1.052
C20:4w6 (Arachidonic)	0.131	0.131	0.132	0.100	0.110
C22:6w3 (Docosa-hexaenoico)	0.121	0.120	0.122	0.111	0.110

**Table 5 animals-13-03472-t005:** Dry matter (DM), gross energy (GE), metabolizable energy corrected for nitrogen (AMEn /kcal/kg), and coefficients of metabolizability (CM%) of lipid sources of plant and animal origin in the diet of European quail.

Variables	Lipid Sources					SEM	*p*-Value
	Soy	Common Maize	Distilled Corn	Poultry Fat	Beef Tallow		
DM Oil (%) *	99.80	99.80	99.60	99.65	99.50	-	-
GE Oil (kcal/kg) *	9746.00	9437.00	9710.00	9809.00	9720.00	-	-
AMEn Oil (kcal/kg)	8554.00	7701.00	7937.00	7906.00	7776.00	1.173	0.2446
MC AMEn Oil (%)	88.01	79.01	84.10	81.43	79.28	1.355	0.1869

SEM: standard error of the mean. * Descriptive analysis.

**Table 6 animals-13-03472-t006:** Coefficients of metabolizability (CM%) of dry matter (DM), ether extract (EE), crude protein (CP), metabolizable energy corrected for nitrogen (AMEn /kcal/kg), and mineral matter (MM) of the test diets with the inclusion of lipid sources of plant and animal origin in the diet of European quail.

Variables	Lipid Sources					SEM	*p*-Value
	Soy	Corn	Distilled Corn	Poultry Fat	Beef Tallow		
CM DM (%)	73.05	73.22	73.08	74.03	72.66	0.797	0.8100
CM EE (%)	81.28 b	83.19 ab	84.04 ab	83.28 ab	84.49 a	0.756	0.0306
CM CP (%)	88.69 b	90.24 ab	90.29 ab	90.94 a	89.73 ab	0.476	0.0247
AMEn in Diet (kcal/kg)	3402.82	3334.51	3353.43	3350.99	3340.57	1.726	0.2446
CM AME in Diet (%)	77.99	76.74	77.32	76.83	76.14	0.522	0.1562
CM MM (%)	22.67	23.85	28.08	28.63	25.47	1.777	0.093

SEM: Standard error of the mean. Averages followed by different letters on the line differ by Tukey’s test at the level of 5% probability.

**Table 7 animals-13-03472-t007:** Performance of 7–21 and 22–35 day phases of European quail fed with lipid sources of plant and animal origin in the diet.

Variables	Lipid Sources					SEM	*p*-Value
	Soy	Corn	Distilled Maize	Poultry Fat	Beef Tallow		
7 to 21 days							
FI bird/day	15.79	15.85	15.73	15.88	15.89	0.166	0.960
FCR	2.58	2.44	2.49	2.53	2.69	0.078	0.170
WG bird/day	6.17	6.52	6.34	6.30	5.93	0.203	0.340
Live weight (g)	138.88	144.00	141.34	140.87	133.65	3.004	0.176
Viability (%)*	100.00	100.00	100.00	100.00	100.00	-	-
22 to 35 days							
FI bird/day	23.15	22.76	23.01	22.94	23.25	0.314	0.850
FCR	2.56	2.78	2.62	2.59	2.47	0.102	0.320
GP bird/day	9.13	8.33	8.97	8.90	9.20	0.330	0.370
Live weight (g)	242.93	234.68	242.70	241.31	237.37	15.511	0.481
Viability (%) *	98.00	99.00	98.00	98.00	99.00	-	-

FI: Feed Intake; FCR: Feed conversion; WG: weight gain. SEM: Standard error of the mean. * Descriptive analysis.

**Table 8 animals-13-03472-t008:** Weight and yield of carcass and cuts of European quail fed with lipid sources of plant and animal origin in the diet.

Variables	Lipid Sources					SEM	*p*-Value
	Soy	Corn	Distilled Maize	Poultry Fat	Beef Tallow		
Housing (g)	171.43	173.10	177.00	179.75	173.25	2.679	0.501
Chest (g)	64.45	66.30	63.40	66.75	61.65	2.539	0.603
Thighs (g)	38.06	38.75	37.55	40.10	36.15	1.393	0.362
Wings (g)	13.86	14.05	14.50	15.40	13.70	0.610	0.290
Back (g)	52.36	53.05	51.50	54.35	51.50	2.147	0.871
Carcass yield (%)	78.33 ab	78.79 ab	81.99 a	77.15 b	78.49 ab	1.068	0.040
Chest (%)	29.63	30.10	28.98	29.58	28.28	1.065	0.781
Thighs (%)	17.51	17.60	17.21	17.82	16.31	0.610	0.670
Wings (%)	6.37	6.39	6.66	6.84	6.29	0.274	0.581
Back (%)	24.09	24.08	23.56	24.15	23.63	0.903	0.984

SEM: standard error of the mean. Averages followed by different letters on the line differ by Tukey’s test at the level of 5% probability.

**Table 9 animals-13-03472-t009:** Weight and yield of viscera and gut size of European quail fed of lipid sources of plant and animal origin in the diet.

Variables	Lipid Sources					SEM	*p*-Value
	Soy	Corn	Distilled Maize	Poultry Fat	Beef Tallow		
Liver (g)	5.06	4.91	4.45	5.38	4.79	0.327	0.36
Heart (g)	1.90	1.96	1.91	1.91	1.90	0.078	0.98
Gizzard (g)	4.92	4.72	4.44	5.17	4.45	0.219	0.10
Spleen (g)	0.12	0.12	0.10	0.12	0.11	0.012	0.62
Intestine (g)	6.47	6.19	6.38	6.72	6.47	0.221	0.56
Weight of reproductive organs (g)	1.88	1.24	1.53	2.13	1.26	0.415	0.68
Liver (%)	2.30	2.20	2.03	2.36	2.19	0.136	0.51
Heart (%)	0.87	0.89	0.88	0.85	0.87	0.035	0.93
Gizzard (%)	2.27	2.14	2.05	2.29	2.04	0.097	0.22
Spleen (%)	0.05	0.05	0.04	0.05	0.05	0.005	0.76
Intestine (%)	2.96	2.80	2.93	2.96	2.95	0.089	0.66
Intestine (cm)	51.18	50.65	49.65	55.10	50.45	2.305	0.49

SEM: standard error of the mean.

**Table 10 animals-13-03472-t010:** Colorimetry and meat quality of European quail fed with lipid sources of plant and animal origin in the diet.

Variables	Lipid Sources					SEM	*p*-Value
	Soy	Corn	Distilled Maize	Poultry Fat	Beef Tallow		
FC (kgf)	0.656	0.692	0.696	0.691	0.66	0.051	0.9679
CL (g)	5.113	5.947	5.865	6.194	6.037	0.299	0.1052
WRC	2.040	2.081	2.583	2.230	1.980	0.318	0.681
CL (%)	9.038	10.173	9.844	10.468	10.627	0.468	0.1341
pH	5.699	5.680	5.729	5.625	5.710	0.0299	0.1423
Colorimetry of the skin of the bird’s chest							
L	61.280	61.930	59.440	61.880	61.900	0.684	0.0428
A	13.230 b	12.800 b	15.570 a	12.040 b	12.390 a	0.392	0.015
B	21.490 a	21.350 a	20.990 a	19.770 b	21.150 a	0.333	0.0019
Colorimetry of the bird’s chest muscle							
L	44.23 a	46.72 a	43.32 b	46.830 a	44.230 b	0.269	0.001
A	12.73 a	13.29 ab	13.49 ab	12.940 b	14.530 a	0.146	0.002
B	14.37 b	14.38 b	14.24 b	15.33 a	14.55 b	0.121	0.003

Averages followed by different letters on the line differ—if by Tukey’s test at the level of 5% probability. L: black/white luminosity; A: Green/red color intensity; B: Blue/yellow color intensity. FC: Shear force; CL: cooking loss; WRC: Water retention capacity; pH: Hydrogen potential; SEM: Standard error of the mean.

## Data Availability

The authors make the data available upon request to the corresponding author.

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
