# Peer review of "Nutritional, Productive, and Qualitative Characteristics of European Quails Fed with Diets Containing Lipid Sources of Plant and Animal Origin"

_animals, 2023, doi:10.3390/ani13223472_

Round 1
Reviewer 1 Report
Notes:
1. In abstract repeating thoughts;
2. Coturnix coturnix coturnix ? lines 169-170;
3. In table 4, more detailed fatty acid profile indicators are needed;
4. In table 3, oil is listed among the ingredients. Is poultry fat and beef tallow oil?
5. What is CM Oil in table 5?
6. Why is quail live weight not provided at the end of the test (table7).
7. Invalid conclusions of the article.
Author Response
Response to Reviewer 1 Comments
I am writing to submit for reviews our research article entitled "Nutritional, productive and qualitative characteristics of European quail fed with unconventional lipid sources for publication in the journal Animals.
- In abstract repeating thoughts;
Changed.
- Coturnix coturnix coturnix ? lines 169-170;
The scientific name "Coturnix coturnix coturnix" refers to the European quail. It follows the binomial nomenclature system, where the first term represents the genus and the second term represents the species. The third "coturnix" in the name represents a specific subspecies of the European quail. In summary, "Coturnix coturnix coturnix" is the scientific name for the European quail, indicating its genus, species, and specific subspecies.
- In table 4, more detailed fatty acid profile indicators are needed;
Changed.
- In table 3, oil is listed among the ingredients.Is poultry fat and beef tallow oil?
Yes, all 5 lipid sources are listed in table 3. Soy, Corn, Distilled maize, Poultry fat, Beef tallow.
- What is CM Oil in table 5?
The correct is dry matter (DM) of the oil
- Why is quail live weight not provided at the end of the test (table7).
The values of final weight of the phases of 7 - 21 days and 22 to 35 days were inserted.
- Invalid conclusions of the article.
Changed.

Reviewer 2 Report
Authors decided to test different fat sources and conclude upon acceptability of use of unconventional types of fats and their energy content.
However, they used normal fats that can be used so title is not correct.
Second, they did not use correct references (tables and top data recources of feed evaluations)
They did not analyse fats and fatty acid profiles. Althouth, they used bomb calorimetry to determine energy content they did not analyse content for moisture and dry matter in order to equalise content of fecal protein, fat and energy.
They did not provide cost evaluation
Conclusions must be written again, to explain the findings.
Minor point, what is corn bran that is written in feed table? if not corn, its chemical composition must be provided- along with all dietary materials, to test to feed composition.
Author Response
Response to Reviewer 2 Comments
- Authors decided to test different fat sources and conclude upon acceptability of use of unconventional types of fats and their energy content. However, they used normal fats that can be used so title is not correct.
Moved as per the guidelines.
- Second, they did not use correct references (tables and top data recources of feed evaluations)
References of national methodologies already validated and used worldwide were used both for the calculation of diets and for the bromatological analysis of foods.
They did not analyse fats and fatty acid profiles. Althouth, they used bomb calorimetry to determine energy content they did not analyse content for moisture and dry matter in order to equalise content of fecal protein, fat and energy.
- Silva, J. H. V. & Costa, F. (2009). Tabelas para codornas japonesas e européias: Tópicos especiais, composição de alimentos e exigências nutricionais Jaboticabal. SP: Funep.
- Silva, D. J.; Queiroz, A. C. Análise de alimentos: métodos químicos e biológicos. 3. ed. Viçosa, MG: Universidade Federal de Viçosa, 2002. 165 p. DOI: http://dx.doi.org/10.5935/2176-4158/rcpa.v21n2p63-67
- Rostagno, H. S., Albino, L. T., Hannas, M. I., Donzele, J. L., Sakomura, N. K., Perazzo, F. G., ... & Brito, C. O. (2017). Brazilian tables for poultry and swine. Feedstuff composition and nutritional requirements. Viçosa, minas gerais. 4th ed. Brazil: Universidade Federal de Viçosa.
- They did not analyse fats and fatty acid profiles. Althouth, they used bomb calorimetry to determine energy content they did not analyse content for moisture and dry matter in order to equalise content of fecal protein, fat and energy.
Dry matter analyses of both oils and diets were performed as indicated in tables 5 and 6.
- They did not provide cost evaluation.
The objective of the authors was not to evaluate the productive costs with the different lipid sources, but to characterize the nutritional values of these foods as well as their influence on the performance and quality of poultry meat. As future evaluations we will provide the cost assessment.
- Conclusions must be written again, to explain the findings.
Changed.
- Minor point, what is corn bran that is written in feed table? if not corn, its chemical composition must be provided- along with all dietary materials, to test to feed composition.
Adjusted, yes it's corn.

Round 2
Reviewer 1 Report
I have no comments.
Author Response
According to the suggestions of the reviewers, new citations were added to improve the text. In addition to an adjustment in completion. In the supplementary files is attached the certificate of translation by company and the document of the ethics committee of the use of animals.

Reviewer 2 Report
Authors have revised adequately the requested points. Only some verey recent references canbe added to provide some very modern views on the subject.
Author Response
According to the suggestions of the reviewers, new citations were added to improve the text. In addition to an adjustment in completion.
